# Increases in Cellular Immune Responses Due to Positive Effect of CVC1302-Induced Lysosomal Escape in Mice

**DOI:** 10.3390/vaccines11111718

**Published:** 2023-11-14

**Authors:** Xiaoming Yu, Yuanyuan Zhang, Liting Hou, Xuwen Qiao, Yuanpeng Zhang, Haiwei Cheng, Haiyan Lu, Jin Chen, Luping Du, Qisheng Zheng, Jibo Hou, Guangzhi Tong

**Affiliations:** 1Institute of Veterinary Immunology & Engineering, Jiangsu Academy of Agricultural Sciences, Nanjing 210014, China; 2National Research Center of Engineering and Technology for Veterinary Biologicals, Jiangsu Academy of Agricultural Sciences, Nanjing 210014, China; 3Jiangsu Key Laboratory for Food Quality and Safety—State Key Laboratory Cultivation Base, Ministry of Science and Technology, Nanjing 210014, China; 4Guo Tai (Taizhou) Center of Technology Innovation for Veterinary Biologicals, Taizhou 225300, China; 5Shanghai Veterinary Research Institute, Chinese Academy of Agricultural Sciences, Shanghai 200241, China

**Keywords:** cross-presentation, cellular immunity, cytosolic pathways, lysosomal escape

## Abstract

This study found a higher percentage of CD8^+^ T cells in piglets immunized with a CVC1302-adjuvanted inactivated foot-and-mouth disease virus (FMDV) vaccine. We wondered whether the CVC1302-adjuvanted inactivated FMDV vaccine promoted cellular immunity by promoting the antigen cross-presentation efficiency of ovalbumin (OVA) through dendritic cells (DCs), mainly via cytosolic pathways. This was demonstrated by the enhanced levels of lysosomal escape of OVA in the DCs loaded with OVA and CVC1302. The higher levels of ROS and significantly enhanced elevated lysosomal pH levels in the DCs facilitated the lysosomal escape of OVA. Significantly enhanced CTL activity levels was observed in the mice immunized with OVA-CVC1302. Overall, CVC1302 increased the cross-presentation of exogenous antigens and the cross-priming of CD8^+^ T cells by alkalizing the lysosomal pH and facilitating the lysosomal escape of antigens. These studies shed new light on the development of immunopotentiators to improve cellular immunity induced by vaccines.

## 1. Introduction

Ideally, adjuvants should improve the humoral and cellular immunity induced by vaccines. An ideal adjuvant should facilitate antigen uptake via antigen-presenting cells (APCs), protect antigens against degradation, and induce strong immune responses. Establishing strong cellular immunity is pivotal for the clearance of viral infections, reduction of viral shedding, and immune surveillance of tumors. The cross-priming of CD8^+^ T cells requires the cross-presentation of antigens on major histocompatibility complex (MHC) class I molecules via antigen-presenting cells (APCs) [1].

Among the APCs, dendritic cells (DCs) are the dominant cell type with highly efficient antigen cross-presentation [2]. Murine DCs can be divided into three main subtypes: conventional DCs (cDCs), plasmacytoid DCs (pDC), and monocyte-derived DCs. The cDCs were further classified as cDC1 (CD8α^+^ and CD103^+^) and cDC2 (CD11b^+^ and CD172α^+^). In general, CD8α^+^ DCs are considered to be the most effective at cross-presentation, irrespective of the antigen type [3]. Different DC subset development is dependent on key cytokines [4], such as granulocyte macrophage-colony stimulatory factor (GM-CSF; Csf-2) [5] and Fms-Related Tyrosine Kinase 3-Ligand (Flt3-L) cytokines [6,7,8]. In this study, we utilized Flt3-L to culture bone marrow cells in order to harvest cDC1 and analyze the CVC1302 mechanism of inducing the cross-presentation of the model antigen OVA.

There are two main pathways for the cross-presentation of exogenous antigens: the vacuolar and the cytosolic pathways [9,10]. Through the vacuolar pathway, antigens are processed by lysosomal proteases and presented to MHCI molecules in the endosomal compartment [11]. Through the cytosolic pathway, antigens are degraded by proteasomes following lysosomal escape and then transported into the endoplasmic reticulum (ER) or endosome via transporters associated with antigen processing (TAP) for loading onto MHCI molecules [11]. Recent studies have clarified that the NADPH oxidase complex (NOX2) assembles upon activation [12] and generates reactive oxygen species (ROS) in the phagosomes, resulting in a phagosome alkalescent environment that facilitates the lysosomal escape of antigens in order to delay antigen degradation and guarantee a sufficient array of antigens for presentation [13,14,15]. It has been well documented that p38 MAPK activation leads to increased phosphorylation of p47phox, which then enhances NOX2 expression and cross-presentation of antigens in the CD8^+^ cDCs [16]. Moreover, it has been demonstrated that p38α plays an important role in the production of IL-12 by CD8^+^ cDCs, which could improve CD8^+^ T cell proliferation in antiviral CTL responses [17,18]. It has been speculated that the TLR agonist-induced activation of p38 might participate in the regulation of DC antigen cross-presentation [19].

We hypothesized that the immunopotentiator CVC1302 activates p38 MAPK signaling and then enhances the cross-presentation of antigens by DC and promotes cellular immune responses through the cytosolic pathway. In this study, we used OVA as a model antigen to test our hypothesis, which could provide a blueprint for designing new generations of immunopotentiators to enhance the CTL responses against viral infections and cancer.

## 2. Materials and Methods

The study protocol was approved by the Science and Technology Agency of Jiangsu Province and the Jiangsu Academy of Agricultural Sciences Experimental Animal Ethics Committee. All efforts were made to minimize animal suffering. All animal studies were performed in strict accordance with the guidelines outlined in the Jiangsu Province Animal Regulations (Government Decree no. 45).

### 2.1. Mice and Piglets

Five-week-old female pathogen-free C57BL/6J mice were purchased from the Yangzhou University (Yangzhou, China). Eight-week-old piglets ((Large White × Landrace) × Duroc), confirmed to be FMDV-negative and seronegative (LPB ELISA titer <1:8), were purchased from Danyang Yunli Animal Husbandry.

### 2.2. Antigen, Adjuvant, and Immunizations

The model antigens, OVA and OVA-FITC, were purchased from Solarbio (Beijing, China). CVC1302 consists of MDP, MPL, and β-glucan, all of which were purchased from InvivoGen (San Diego, CA, USA). All the components were dissolved in sterile water at the appropriate concentrations of 25 μg/mL. For mouse immunization, CVC1302 was mixed with OVA at a ratio of 1:2 and then emulsified with ISA206 at a ratio of 1:1. This was designated OVA-CVC1302. OVA mixed directly with ISA206 at a ratio of 1:1 was designated OVA. For piglet immunization, aqueous-phase CVC1302 was emulsified with ISA206 and then fully mixed with an FMDV-killed vaccine at a ratio of 1:9, which was designated as FMDV-CVC1302.

The mice were intramuscularly immunized with OVA-CVC1302 or OVA alone, with 50 μg OVA for each mouse or 1 μL CVC1302 for each mouse.

The piglets were immunized intramuscularly with 2 mL FMDV-CVC1302 vaccine or FMDV vaccine alone.

### 2.3. CD3^+^CD8^+^ T Cells Differentiation in Mice and Piglets

Inguinal lymph nodes were collected from the immunized mice at 7 days post-immunization (dpi), and single lymphocytes were harvested as described previously [20]. Cells were stained for 30 min at 4 °C with the following mAbs: CD3-FITC, CD4-Percp-cy5.5 and CD8-APC. Heparinized blood samples were sampled from the immunized piglets at 28 dpi. PBMCs were harvested as described previously [21]. Single cells were incubated with CD3-FITC, CD4-Percp-cy5.5 and CD8-APC. The cells were acquired using a BD Accuri C6. The data were analyzed using FlowJo version 7.6.1 software.

### 2.4. Cytotoxicity Assay

The cytotoxicity assays were performed as previously described [22]. In brief, the splenocytes obtained from naïve mice were divided into two groups, followed by stained with SIINFEKL peptide-pulsed 2.5 μM CFSEhigh or 0.25 μM CFSElow alone, respectively. CFSEhigh and CFSElow cells were mixed at a 1:1 ratio. Then, 1 × 10^7^ cells in 200 μL of RPMI-1640 were adoptively transferred into the groups of mice immunized with OVA-CVC1302, OVA, and PBS, as well as a blank group. After 20 h, spleen and inguinal lymph nodes were sampled after twenty hours. The single cells were prepared and analyzed using flow cytometry. The killing rate was analyzed using the following formula [23]:Killing rate = 100% − [(%CFSEhigh immunized mouse/% CFSElow immunized mouse)/(%CFSEhigh naïve mouse/% CFSElow naïve mouse) × 100%]

### 2.5. Preparation of Bone Marrow-Derived DCs (BMDCs)

The BMDCs were cultured using Flt3-L and were prepared as described previously [24]. The BMDCs were harvested at 7 days to analyze the ability of CVC1302 to induce T cell proliferation.

### 2.6. CCK-8 Assay

The cytotoxicity of CVC1302 on the BMDCs was analyzed using the CCK-8 assay [25]. Different concentrations of OVA and CVC1302 were added into 96-well plates, which were seeded with BMDCs at a density of 1 × 10^5^ cells/well. After co-culturing for 12 h, 10 μL of CCK-8 solution was added to each well. Two hours later, absorbency was measured at 450 nm. The cell viability was calculated using the following formula:Cell viability = ([OD_experimental samples_ − OD_controls_]/[OD_untreated samples_ − OD_controls_]) × 100

### 2.7. BMDCs Activation

Immature BMDCs were plated at 5 × 10^5^ cells/well in 24-well plates and then co-cultured with OVA, OVA plus CVC1302, or PBS for 6 h at 37 °C with 5% CO_2_. The DCs were collected and stained for 30 min at 4 °C with the following mAbs: anti-CD11c FITC, anti-CD40 APC, anti-CD80 APC, anti-CD86 APC, anti-MHC I APC, and anti-MHC II APC. The cells were analyzed using the BD Accuri C6. Data analyses were performed using the FlowJo version 7.6.1 software.

### 2.8. Cytokine Detection from Cell Culture Supernatants

The supernatants from the cultures of BMDCs were treated with OVA with or without CVC1302 for 6 h. The cytokines in the cell culture supernatants were quantified using an IL-12p70 ELISA kit, according to the manufacturer’s protocol.

### 2.9. B3Z Cross-Presentation Assays

The ability of CVC1302 to induce cross-presentation was analyzed using B3Z T cells [26].

Flt3-L-cultivated BMDCs and OVA or OVA+CVC1302 were co-cultured for 16 h at 37 °C with 5% CO_2_ in 96-well plates. Following this, the B3Z cells were added for 24 h, and buffer Z (100 mM 2-mercaptoethanol, 9 mM MgCl_2_, 0.125% NP-40, and 0.15 mM chlorophenol red β–galactoside) were added and incubated for another 6 h. Then, stop buffer (300 mM glycine and 15 mM EDTA in water) was added, and the absorbency at 570 nm was measured. The B3Z cell activation data are shown as the normalized optical density (OD) relative to the control group.

### 2.10. Confocal Microscopy

The cells from bone marrow were seeded in 6-well plates at a density of 2 × 10^6^ cells/well on 1.5 mm microscope slides and treated with Flt3-L 7 days at 37 °C with 5% CO_2_. At day 7, the coverslips were removed into a 24-well plate following washing with PBS, and then LysoTracker^®^ Red was added for 30 min at 37 °C with 5% CO_2_ in the dark. After washing three times, OVA-FITC, OVA-FITC+CVC1302, or PBS was added into the wells for 40 or 60 min at 37 °C with 5% CO_2_. BMDCs were washed with PBS and fixed with periodate-lysine-paraformaldehyde (PLP, Solarbio, Beijing, China). DAPI was used to stain the nuclei. The sections were visualized and photographed using a Zeiss LSM700 confocal microscope (objective 320). Images were acquired using Zeiss LSM image browser software 2.4 (Zeiss, Oberkochen, Germany).

### 2.11. Real-Time PCR for Gene Expression

The total RNAs from BMDCs treated with OVA or OVA+CVC1302 were extracted using Trizol and reverse transcribed in a 20 μL reaction mixture. The resulting cDNA product was amplified in a 20 μL real-time PCR reaction. The primers used for the mouse NOX2 gene were 5′-TGTGGTTGGGGCTGAATGTC-3′ and 5′-CTGAGAAAGGAGAGCAGATTTCG-3′. The primers used for the mouse GAPDH gene were 5′-TGTGTCCGTCGTGGATCTGA-3′ and 5′-CCTGCTTCACCACCTTCTTGA-3′.

### 2.12. Immunoblot

The BMDCs treated with OVA or OVA+ CVC1302 were harvested and lysed with a RIPA buffer (CST) containing a protease and phosphatase inhibitor cocktail. Western blot analysis was performed as described previously [17]. The sample protein was loaded and resolved on a 12% SDS-PAGE gel and transferred to the PVDF membrane. After blocked with 5% BSA in PBST for phosphor-protein and 5% non-fat milk for total protein in PBST for 1 h, the membranes were probed with anti-total p38α, anti-phosphor-p38α-T180/Y182, anti-NOX2/gp91phox, and anti-β-actin primary antibodies. Then, the membranes were extensively washed with PBST, followed by incubation with goat anti-rabbit IgG-HRP antibodies diluted in 5% non-fat milk in PBST for 1 h at 37 °C. Protein bands were visualized using the ECL prime western blotting detection reagent. The relative expression levels of target proteins according to the grayscale of β-actin or the indicator proteins in cells were assayed using ImageJ 10.8.1 software, as described previously [17].

### 2.13. ROS Formation—DCF Assay

The production of ROS was quantified using the DCF assay as described previously [27]. In brief, BMDCs were seeded into 96-well plates at a density of 2 × 10^6^ cells/well, and then 2′,7′-dichlorofluorescin diacetate (DCFH-DA) was added at a terminal concentration of 10 μM for 20 min at 37 °C with 5% CO_2_ in the dark. After washing with PBS, OVA or OVA+CVC1302 was added for 1 or 3 h at 37 °C with 5% CO_2_ in the dark. The fluorescence intensity was measured in 15 min intervals up to 3 h at Ex/Em of λ = 485/535 nm (Synergy 2, Biotek Instruments GmbH, Bad Friedrichshall, Germany).

### 2.14. AO Staining of BMDCs

AO staining was performed as described previously [28,29]. Briefly, BMDCs were treated with OVA or OVA+CVC1302 for 4 h in a 96-well plate. Then, AO (2 μg/mL) was added for 15 min at 37 °C. The fluorescence signal of AO (excitation: 490 nm; emission: 650 nm) in the lysosomal compartments was analyzed via flow cytometric analysis.

### 2.15. LysoSensor Fluorescence Assay

The effect of CVC1302 on the lysosomal pH was analyzed using the LysoSensor fluorescence assay as described previously [30,31]. BMDCs were treated with OVA or OVA+CVC1302 for 4 h in a 96-well plate. LysoSensor Green DND-189 dye (1 μM in a pre-warmed medium) was added at 37 °C for 30 min. The LysoSensor fluorescence signal (excitation: 443 nm; emission: 505 nm) was measured via flow cytometric analysis.

### 2.16. Statistical Analysis

Statistical analyses were performed using GraphPad Prism version 5 (GraphPad 9 Software, San Diego, CA, USA). The differences among groups were assessed using a one-way analysis of variance (ANOVA), followed by Tukey’s post-hoc *t*-test. The differences between groups were assessed using the Student’s *t*-test. Statistical significance was set at *p* < 0.05. All data shown in the manuscript are expressed as the mean ± standard error of the mean (SEM).

## 3. Results

### 3.1. CVC1302 Leads to an Enhanced CTL Response

In order to analyze the potential of CVC1302 to induce a CTL response, we immunized the mice with OVA adjuvanted with or without CVC1302, emulsified with ISA206. The Drained lymph nodes were sampled at 7 dpi, and the CD3^+^ CD8^+^ T cells were analyzed via flow cytometry. As shown in Figure 1A,B, there were significantly higher percentages of CD3^+^ CD8^+^ T cells in the mice immunized with OVA-CVC1302 than in those immunized with OVA. Meanwhile, we detected the percentage of CD3^+^ CD8^+^ T cells in the piglets immunized with FMDV-CVC1302. Similar to mice, a higher percentage of CD3^+^ CD8^+^ T cells was also observed in piglets immunized with FMDV-CVC1302 than in those immunized with FMDV (Figure 1C,D). Furthermore, we found no significant difference in CD4^+^ CD8^+^ T cells between mice immunized with OVA and pigs immunized with FMDV.

To investigate the effect of CVC1302 on cytotoxic T cell responses, we performed an in vivo CTL killing assay using the spleen and draining lymph nodes against the target cells loaded in vitro with SIINFEKL. The results showed that the killing rates for SIINFEKL-loaded target cells at 7 dpi were doubled compared to the killing rates observed in those immunized with OVA not only in the spleen (Figure 1E, F) but also in the draining lymph nodes (Figure 1G,H) of the OVA-CVC1302-immunized mice.

### 3.2. CVC1302 Induces Improved Cross-Presentation by Flt3-L-Cultured BMDCs

The impact of CVC1302 cytotoxicity was assessed by determining the viability of the DCs after 12 h. As shown in Figure 2A, the Flt3-L-cultured BMDCs were cultivated with OVA at a 20 μg/mL concentration with or without CVC1302 at 0, 5, 10, 50, and 100 μg/mL concentrations. There were no significant differences in BMDCs viability between the OVA- or OVA+CVC1302- treated BMDCs at any concentration, which indicates that CVC1302 has good compatibility with BMDCs and could be an added immunopotentiators added to vaccines.

As demonstrated, mature DCs have an enhanced potency of antigen presentation capacity in order to activate T cells; we observed the ability of CVC1302 to induce DC maturation. As shown in Figure 2B, OVA+CVC1302 improved the expression levels of CD40, CD80, CD86, and MHC I in BMDCs compared with OVA alone. These results indicated that CVC1302 induced the maturation of BMDCs to improve the potency of antigen presentation.

The ability of CVC1302 to improve the cross-presentation of BMDCs was detected using B3Z cells. As shown in Figure 2C, following 24 h of incubating the BMDCs with B3Z cells, there was a more significant increase in T cell activation for the OVA+CVC1302 group compared to that of OVA of ~ one-fold.

Interleukin-12 (IL-12) produced by APCs acts as a third signal for CTL activation and enhances CTL proliferation and effector functions [32]. In this study, the expression levels of IL-12 by OVA+CVC1320-cultivated BMDCs were significantly higher than OVA of 2.5 fold (*p* < 0.0001), which means that CVC1302 could enhance the expression of IL-12 by BMDCs in order to promote CTL proliferation (Figure 2D).

Together, these data show that CVC1302 promotes the maturation of BMDCs and then enhances the cross-presentation of BMDCs and the expression levels of IL-12 by BMDCs, ultimately improving the cross-priming and proliferation of T cells.

### 3.3. CVC1302-Induced Lysosomal Escape of Antigen Is Indispensable for Cross-Presentation

Cross-presentation pathways include the vacuolar and cytosolic pathways. To unravel the underlying mechanisms of CVC1302 in improving the efficiency of cross-presentation by BMDCs, the main cross-presentation pathway was clarified using confocal microscopy. The microscopic images (Figure 3) showed that the OVA+CVC1302-treated BMDCs harbored many more pseudopods or dendrites compared to those of the OVA-treated BMDCs. Furthermore, the MOI of green fluorescence (antigen (OVA-FITC, green)) was significantly higher in the OVA+CVC1302-treated BMDCs when compared to that of the OVA-treated BMDCs. In the OVA+CVC1302-treated BMDC_S_, lots of green fluorescence and red fluorescence (lysosome) were un-co-located at 40- and 60-min post-cultivation; however, green fluorescence and red fluorescence were co-located in the OVA-treated BMDCs. These data show that CVC1302 induces maturation by observing the morphology and mediates the cross-presentation via the lysosomal escape of antigens.

### 3.4. CVC1302 Mediates Lysosomal Escape via p38α Signaling

It was demonstrated that the DCs activated by TLR agonists through p38α MAPK signaling induce NOX2 production and generate ROS, causing the alkalinization of lysosomes and finally facilitating the lysosomal escape of antigens.

First, we observed that the phosphorylated level of p38α in the OVA+CVC1302-treated BMDCs was much higher than that in OVA-treated BMDCs, as demonstrated by western blotting (Figure 4A).

Second, the results of flow cytometry showed that the OVA+CVC1302-treated BMDCs produced higher levels of NOX2 than the OVA-treated BMDCs (Figure 4B). According to the above conclusions, we speculated that the activation of NOX2 by CVC1302 is related to the p38α MAPK signaling; however, whether p38α MAPK signaling is the only factor causing the production of NOX2 induced by CVC1302 requires further research using p38α-deficient DCs.

Third, ROS production in the OVA+CVC1302-treated BMDCs was detected using a DCFH-DA probe. As shown in Figure 4C, the cellular ROS levels in BMDCs were significantly higher within both 1 and 3 h after treatment with OVA+CVC1302 compared to those of OVA alone.

Finally, we studied whether the increased production of ROS modulated the intracellular acidity in the OVA+CVC1302-treated BMDCs. Two lysosomal pH indicators, AO and LysoSensor dye, were used to evaluate the lysosomal pH in the BMDCs after the OVA+CVC1302 treatment. Concomitant with ROS production, the BMDCs pulsed with OVA+CVC1302 exhibited a significantly decreased fluorescence intensity of AO and LysoSensor dye compared to that of the OVA-treated BMDCs (Figure 4D,E).

Taken together, these results suggested that OVA+CVC1302 leads to the alkalization of lysosomes in BMDCs via the production of NOX2 and ROS through p38α MAPK signaling.

## 4. Discussion

CTL immunity is particularly vital for the clearance of viral infections and the eradication of cancer. Improving the efficiency of DC cross-presentation by adding an appropriate adjuvant can trigger potent CTL responses.

The immunopotentiator CVC1302, a complex of pathogen recognition receptors (PRRs), potentiates CD8 T cell responses in mice and piglets. Significantly enhanced B3Z cell activation occurred when they were incubated with the OVA+CVC1302-treated BMDCs, showing that CVC1302 improved the cross-priming of CD8^+^ T cells independent of the enhanced ability of cross-presentation in the OVA+CVC1302-treated BMDCs.

Our study clarified the mechanism of CVC1302 in inducing the improved cross-presentation of DCs derived from Flt3L-BM cultures, which presented the majority of the DCs in the lymphoid tissues, not the GM-CSF-induced MoDCs. This design means that our results derived from the in vitro experiments were similar to those derived from the in vivo tests.

Generally, the two cross-presentation pathways are referred to as “vacuolar” and “cytosolic” [2]. The lysosomal escape of antigens in the OVA+CVC1302-treated BMDCs was observed via confocal microscopy at 40 and 60 min after cultivation, which means that CVC1302 mediated the cross-presentation of antigens via cytosolic pathway. The key steps in the cytosolic pathway include generating an alkaline phagosomal environment, promoting antigen escape from lysosomes, and recruiting the ER to the phagosomes [17]. To further clarify the CVC1302 mechanism by potentiating antigen cross-presentation via lysosomal escape, we detected the production of NOX2 and ROS, which play important roles in elevating the pH levels in phagosomes, leading to phagosomal alkalization. The results showed that OVA plus CVC1302 significantly enhanced the production of NOX2 and ROS in BMDCs compared with OVA alone.

Some studies have reported that the enhanced DC antigen capture ability induced by TLR signaling is mediated by the p38α signaling pathway [19,33]. Moreover, the deletion of p38α results in a reduction of the expression of NOX2 and ROS to impair the antigen cross-presentation by DCs and the suppression of IL-12 expression by cDCs to impair the cross-priming of CD8+ T lymphocytes [16,17]. Significantly higher levels of phosphorylated p38α, NOX2, ROS, and IL-12 in the OVA+CVC1302-treated BMDCs and promoted the CTL responses observed in our study were in line with previous research.

## 5. Conclusions

In summary, our study clarified that the CVC1302 mechanism of inducing promoted CTL responses was independent of the DC’s improved ability for cross-presentation via antigen lysosomal escape, which is mediated through the p38α signaling pathway. Our findings provide new insights into the development of new immunopotentiators to trigger and potentiate CTL responses against infectious diseases and tumors.

## Figures and Tables

**Figure 1 vaccines-11-01718-f001:**
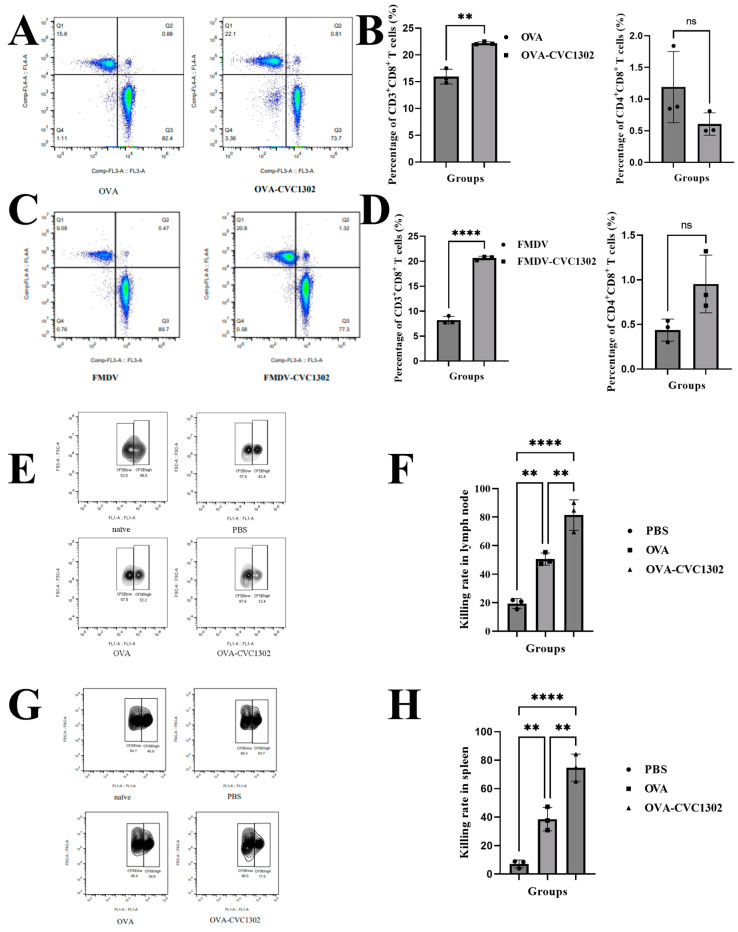
CVC1302 treatment led to an enhanced CTL response. (**A**,**B**) C57BL/6J mice were immunized with OVA or OVA-CVC1302. Inguinal lymph nodes were sampled from mice at 7 dpi and analyzed via flow cytometry. (**A**) Representative flow cytometry plot showing CD3^+^ CD8^+^ T cells. (**B**) Percentages of CD3^+^ CD8^+^ T cells (left) and percentages of CD4^+^ CD8^+^ T cells (right) in mice. (**C**,**D**) Piglets were immunized with FMDV or FMDV-CVC1302. Sera were sampled from piglets at 28 dpi. PBMCs were prepared and analyzed using flow cytometry. (**C**) Representative flow cytometry plot showing CD3^+^ CD8^+^ T cells. (**D**) Percentages of CD3^+^ CD8^+^ T cells (left) and percentages of CD4^+^ CD8^+^ T cells (right) in piglets. (**E**,**H**) SIINFEKL peptide-pulsed target cells were transferred into mice immunized with OVA or OVA-CVC1302 at 7 dpi. The killing rates of the target cells were analyzed using flow cytometry. (**E**) Representative cytometry plot showing the target cells in the lymph nodes of mice immunized with OVA or OVA-CVC1302. (**F**) Killing rates in the lymph nodes of mice immunized with OVA or OVA-CVC1302. (**G**) Representative cytometry plot showing the target cells in the spleens of mice immunized with OVA or OVA-CVC1302. (**H**) Killing rates in the spleens of mice immunized with OVA or OVA-CVC1302. Data are presented as the mean ± SEM; ** *p* ≤ 0.05, **** *p* ≤ 0.0001.

**Figure 2 vaccines-11-01718-f002:**
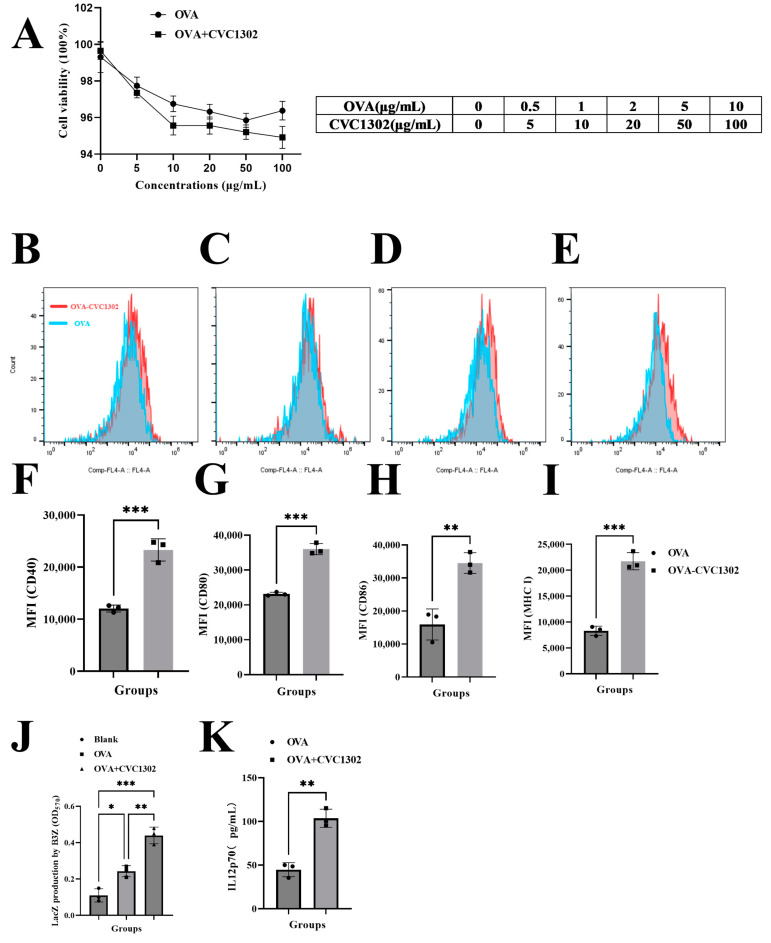
CVC1302 improved antigen cross-presentation by Flt3-L-cultured BMDCs. (**A**) The cytotoxicity of CVC1302 in Flt3-L-cultured BMDCs. A series of concentrations of CVC1302 were incubated with BMDCs for 12 h, and cellular viability was measured using the CCK-8 assay. (**B**–**I**) The activation of BMDCs treated with OVA±CVC1302 for 6 h. The cell surface expression of CD40, CD80, CD86, and MHCI in BMDCs was assessed by flow cytometry. (**B**) Representative cytometry plot showing the expression of CD40 in BMDCs. (**C**) Mean fluorescence intensity of CD40 in the BMDCs. (**D**) Representative cytometry plot showing the expression of CD80 in BMDCs. (**E**) The mean fluorescence intensity of CD80 in BMDCs. (**F**) Representative cytometry plot showing the expression of CD86 in BMDCs. (**G**) The mean fluorescence intensity of CD40 in BMDCs. (**H**) Representative cytometric plot showing the expression of MHCI in BMDCs. (**I**) Mean fluorescence intensity of MHCI in the BMDCs. (**J**) β-galactosidase production by B3Z cells after co-culture with BMDCs pre-treated with OVA±CVC1302 for 24 h. (**K**) The expression levels of IL-12p70 in BMDCs treated with OVA±CVC1302 for 6 h. Data are presented as the mean ± SEM; * *p* ≤ 0.01, ** *p* ≤ 0.01, *** *p* ≤ 0.001.

**Figure 3 vaccines-11-01718-f003:**
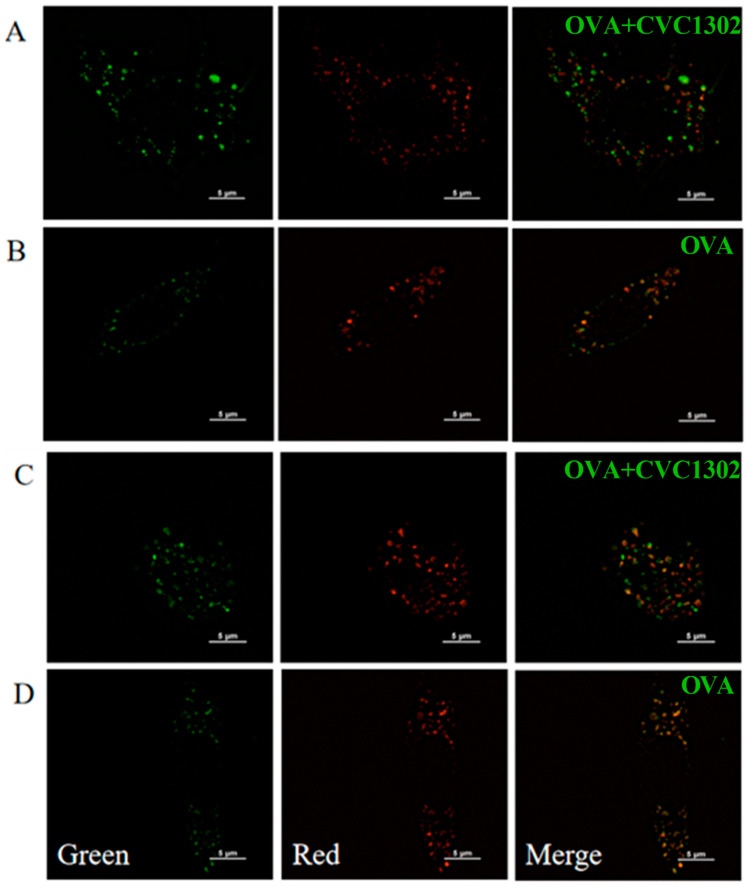
CVC1302 engages in the cytosolic pathway during cross-presentation. BMDCs were pulsed with OVA-FITC±CVC1302, chased at the indicated time points, and immunostained to assess the degree of OVA co-localization with LysoTracker Red DND-99 (lysosome). (**A**) BMDCs were treated with OVA+CVC1302 for 40 min. (**B**) BMDCs were treated with OVA for 40 min. (**C**) BMDCs were treated with OVA+CVC1302 for 60 min. (**D**) BMDCs were treated with OVA for 60 min. Scale bars: 5 μm.

**Figure 4 vaccines-11-01718-f004:**
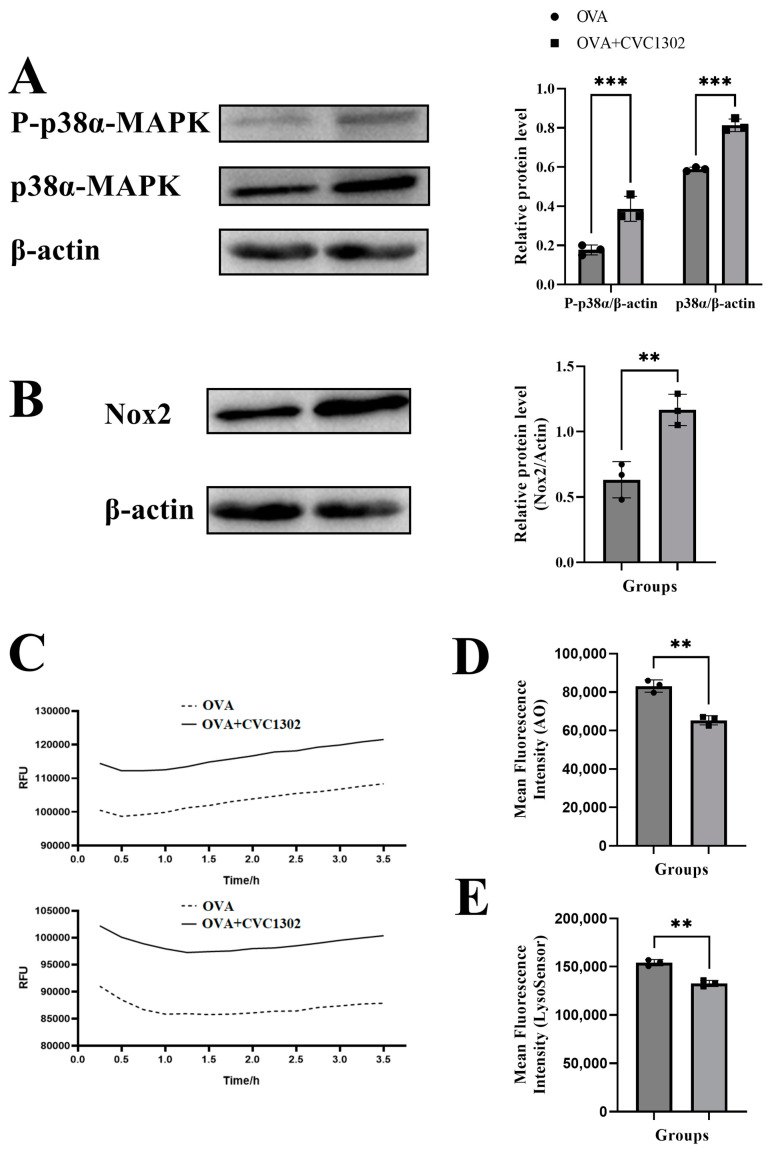
CVC1302 mediates lysosomal escape via p38α signaling. (**A**,**B**) Total proteins were harvested from BMDCs treated with OVA±CVC1302 for 6 h. Next, proteins (**A**) p38α and P-p38α were activated in BMDCs treated with OVA+CVC1302. BMDCs were stimulated with OVA±CVC1302 as indicated. (**B**) NOX2 was activated in BMDCs treated with OVA+CVC1302. (**C**) ROS were activated in BMDCs treated with OVA+CVC1302. (Upper) BMDCs were treated with OVA+CVC1302 for 1 h. (Bottom) BMDCs were treated with OVA+CVC1302 for 3 h. (**D**,**E**) CVC1302 interferes with the acidification of lysosomal pH in BMDCs. After BMDCs were treated with OVA±CVC1302 for 6 h, BMDCs were stained with AO or LysoSensor, followed by the measurement of AO or LysoSensor signal with flow cytometric analysis. (**D**) The MFI of BMDCs stained with AO after OVA±CVC1302 treatment. (**E**) The MFI of BMDCs stained with LysoSensor after OVA±CVC1302 treatment. Data are presented as the mean ± SEM; ** *p* ≤ 0.01, *** *p* ≤ 0.001.

## Data Availability

The data presented in this study are available in this article.

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
