# Peer review of "Increases in Cellular Immune Responses Due to Positive Effect of CVC1302-Induced Lysosomal Escape in Mice"

_vaccines, 2023, doi:10.3390/vaccines11111718_

Round 1

Reviewer 1 Report

Comments and Suggestions for Authors The manuscript Promoted cellular immune responses due to positive effect of CVC1302-induced lysosomal escape in mice It Will be important to better justify the inclusion of tests with cells from pigs, because it is well identified that in this species there are cell groups with the possibility of having a double CD4+CD8+ label, therefore the attempt to associate the results in both species should be better identified and/or discussed. The comment on the L 237 results would be incomplete (as long as both CD4+ and CD8+ are not analyzed in these populations) or without correlation with the data obtained in mice. Some results such as those presented in section 3.2 are interesting, however would it be important to try to explain what cytotoxicity mechanism was activated? Expression and secretion of cytotoxic factors such as perforin or granzyme? It is interesting to identify that the work tries to explain the potential mechanisms of this adjuvant through lysosomal escape in mice using mainly the adjuvant without some type of specific control and on the other hand the use of another species with different cellular phenotypic characteristics. The observed object is due to the synergy of the three main components of this adjuvant? Minor comments Concerning Pigklets is important to indicate which genera and species? Since the pigs as members of the Suidae family, includes 8 genera and 16 species P3 After centrifugation at 1,000 rpm “with” 5 min? is better delete with...

Author Response

The manuscript Promoted cellular immune responses due to positive effect of CVC1302-induced lysosomal escape in mice. It will be important to better justify the inclusion of tests with cells from pigs, because it is well identified that in this species there are cell groups with the possibility of having a double CD4+CD8+ label, therefore the attempt to associate the results in both species should be better identified and/or discussed.

Reply: Thanks for the advice from the reviewer. The CD4+CD8+ T cells had been identified in the revised manuscript as shown in Fig.1A and C and the discussion had been added in the revised manuscript as shown in L218-219.

The comment on the L237 results would be incomplete (as long as both CD4+ and CD8+ are not analyzed in these populations) or without correlation with the data obtained in mice.

Reply: Thanks for the advice from the reviewer. We have re-analyzed the percentages of CD4+ and CD8+ T cells, and the discussion had been rewritten in the revised manuscript as shown in L218-219.

Some results such as those presented in section 3.2 are interesting, however would it be important to try to explain what cytotoxicity mechanism was activated? Expression and secretion of cytotoxic factors such as perforin or granzyme?

Reply: Thanks for the advice from the reviewer. In this research, we had only paid attention on whether CVC1302 had cytotoxicity to BMDC. No more ideas on the mechanism of cytotoxicity induced by CVC1302. According to the suggestion from the reviewer, we will explore the mechanism of cytotoxicity induced by CVC1302 in our further study.

It is interesting to identify that the work tries to explain the potential mechanisms of this adjuvant through lysosomal escape in mice using mainly the adjuvant without some type of specific control and on the other hand the use of another species with different cellular phenotypic characteristics. The observed object is due to the synergy of the three main components of this adjuvant?

Reply: Thanks for the advice from the reviewer. At the beginning of the study, we were not sure the enhanced cross-presentation of CVC1302 relied on the lysosomal escape, so we did not select another specific control. In our future, we will confirm the mechanism of CVC1302 in inducing cross-presentation by using BMDC cultured from piglets. In this research, we concluded that the lysosomal escape induced by CVC1302 is due to the synergy of the three main components, and in the further study by using BMDC cultured from piglets, we will make understood whether the observed object is due to the synergy of the three main components or only one components of CVC1302.

Minor comments Concerning Pigklets is important to indicate which genera and species? Since the pigs as members of the Suidae family, includes 8 genera and 16 species.

Reply: According to the suggestion from the reviewer, we had added the genera and species of piglets in the revised manuscript as shown in L86-87.

After centrifugation at 1,000 rpm “with” 5 min? is better delete with...

Reply: According to the suggestion from the reviewer, the sentence had been rewritten in the revised manuscript.

Reviewer 2 Report

Comments and Suggestions for Authors

In this manuscript, the authors show that higher ROS levels and a noticeable increase in lysosomal pH in DC enhance the lysosomal escape of OVA. Mice immunized with OVA-CVC1302 displayed notably improved CTL activities. Overall, CVC1302 augments the cross-presentation of exogenous antigens and the cross-priming of CD8+ T cells by elevating lysosomal pH, thereby facilitating the lysosomal escape of the antigen. However, the manuscript presents several systemic issues. The quality of the English writing could be improved, abbreviations are not properly defined, and there are multiple typos. Moreover, there are citation errors, such as referencing Figure 1C on line 245, and Figures 1EFGH appear to be overlooked. Additionally, the resolution of Figure 3 is suboptimal. I recommend that the authors refine and restructure the manuscript before resubmitting for further review.

Comments on the Quality of English Language

need to be improved.

Author Response

In this manuscript, the authors show that higher ROS levels and a noticeable increase in lysosomal pH in DC enhance the lysosomal escape of OVA. Mice immunized with OVA-CVC1302 displayed notably improved CTL activities. Overall, CVC1302 augments the cross-presentation of exogenous antigens and the cross-priming of CD8+ T cells by elevating lysosomal pH, thereby facilitating the lysosomal escape of the antigen.

However, the manuscript presents several systemic issues. The quality of the English writing could be improved, abbreviations are not properly defined, and there are multiple typos. Moreover, there are citation errors, such as referencing Figure 1C on line 245, and Figures 1EFGH appear to be overlooked. Additionally, the resolution of Figure 3 is suboptimal. I recommend that the authors refine and restructure the manuscript before resubmitting for further review.

Reply: According to the suggestion from the reviewer, the Figure 1C, E, F, G and H had been corrected in the revised manuscript as shown in L213, 218, 222 and 223. We were so sorry for the resolution of Figure 3 because of the instrument.

Round 2

Reviewer 1 Report

Comments and Suggestions for Authors

The main queries were solved and the manuscript has been improved and the manuscript could be accepted

Author Response

Thanks a lot.

Reviewer 2 Report

Comments and Suggestions for Authors

The manuscript contains many grammatical errors. It would be beneficial to have it reviewed and edited by a native English speaker.

Comments on the Quality of English Language

The manuscript contains many grammatical errors. It would be beneficial to have it reviewed and edited by a native English speaker.
